# Role of serum uric acid in ischemic stroke: A case-control study in Bangladesh

**Mohammad Ibrahim Khalil**[1☯], **Marium Salwa**[2☯]*, **Sarmin Sultana**[2], **Mohammad Abdullah Al Mamun**[3], **Nilima Barman**[4], **M. Atiqul Haque**[2]

**1** Department of Neurology, Shaheed Suhrawardy Medical College Hospital, Dhaka, Bangladesh,
**2** Department of Public Health and Informatics, Bangabandhu Sheikh Mujib Medical University, Dhaka, Bangladesh, **3** Department of Epidemiology & Research, National Heart Foundation Hospital & Research Institute, Dhaka, Bangladesh, **4** Department of Laboratory Medicine, Bangladesh Institute of Research and Rehabilitation for Diabetes, Endocrine and Metabolic Disorders (BIRDEM), Dhaka, Bangladesh

☯ These authors contributed equally to this work.
* mariumsalwa@gmail.com

**Data Availability Statement:** All data files are available from the Mendeley Data repository: https://data.mendeley.com/datasets/z9m5jzcbdb/1. Citation: salwa, marium (2020), "Ishemic stroke

## Abstract

### Introduction

Increased level of serum uric acid (SUA) is often considered a risk factor for ischemic stroke. This study was conducted to examine the association of SUA level with ischemic stroke and assessed gender-based differences, if any.

### Methods

In this case-control study, neuroimaging-confirmed ischemic stroke patients were recruited as cases within three days of an incident from neurology in-patient department, and as controls, patients without stroke history were recruited from neurology out-patient department. Blood was collected from the respondents of both groups to assess SUA level, lipid profile and oral glucose tolerance test. Binary logistic regression was done for estimating the risks of ischemic stroke.

### Results

A total of 338 participants were recruited, where 169 were cases and 169 were controls. Around 60 percent respondents of both case and control groups were male. Mean SUA levels for cases and controls were 6.03 (SD 1.84) mg/dl and 4.04 (SD 1.46) mg/dl, respectively. After adjustment for age, tobacco consumption status, diabetes, hypertension, coronary heart disease and dyslipidemia, elevated SUA level was found to be significantly associated with ischemic stroke only in females (OR = 1.49; 95% CI = 1.01–2.19; p<0.05). Overall, each unit increase in SUA level exhibits 25 percent increment in odds of having ischemic stroke (OR = 1.25; 95% CI = 1.02–1.5372; p<0.05).

and serum uric acid", Mendeley Data, v1 (http://dx.doi.org/10.17632/z9m5jzcbdb.1).

**Funding:** The author(s) received no specific funding for this work.

**Competing interests:** The authors have declared that no competing interests exist.

## Conclusion

This study concluded that elevated SUA level is significantly associated with the acute phase of an ischemic stroke and gender-specific analysis demonstrates this association only in females.

## Introduction

The role of serum uric acid (SUA) in the prediction of ischemic stroke has been a debated issue for the past few decades. A growing body of research suggests that there is a positive association between SUA and ischemic stroke [1,2]. However, elevated SUA level is also found to be associated with a number of common risk factors of ischemic stroke like hypertension [3], obesity [4], dyslipidemia [5], metabolic syndrome [6] and diabetes [7,8]. Therefore, it is difficult to establish an independent relation of hyperuricemia with ischemic stroke. Moreover, some studies revealed hyperuricemia as a protective factor of ischemic stroke, while some other studies suggested uric acid as a part of treatment at the acute phase of stroke [9–11]. These disputes surrounding the role of SUA emphasize the need of more research to explore its association with ischemic stroke considering the independent influence of cardiovascular risk factors.

Stroke is ranked as third in years of life lost [12] and also identified as a leading cause of long-term adult-onset physical disability [13]. Bangladesh, a lower-middle income country of South Asia, has been observing a rise in stroke amongst its population over the last decade with a current prevalence of 1.96 per 1000 population [14]. However, comprehensive population-based data regarding incidence, prevalence and potential risk factors have little been studied. So far, identification of modifiable risk factors and implication of risk reduction strategies remain the mainstay in stroke prevention [15]. In addition to well-established risk factors like hypertension, diabetes, hyperlipidemia, and unhealthy lifestyle, some less-studied factors associated to ischemic stroke like SUA, which can easily be measured and treated, need to be explored to design an appropriate prevention strategy. Hence, the purpose of this study was to evaluate the role of SUA level in ischemic stroke taking the independent effect of other cardiovascular risk factors into consideration, and to assess sex-based differences, if any.

## Research question

1. What is the role of SUA level in acute phase of ischemic stroke while controlling other cardio-metabolic risk factors?

2. Is there any gender-based difference in the role of SUA level in acute phase of ischemic stroke?

## Methodology

### Study design and setting

This case-control study was carried out at the Department of Neurology at Shaheed Suhrawardy Medical College Hospital, a tertiary care teaching hospital in Dhaka, Bangladesh between January to June 2016. More than 2000 stroke patients per year are admitted and cared

for at this hospital. Besides, more than 10,000 patients per year are cared for different neurological complaints at out-patient services. All stroke patients admitted in in-patient department and all patient with other neurological complaints attended in out-patient department of Neurology were regarded as the study population.

## Cases and controls

Cases were recruited consecutively from patients who were admitted within three days of an acute ischemic stroke in the in-patient department and aged 40 years or above. Ischemic stroke was confirmed by computed tomography scan and/or magnetic resonance imaging of the brain. After confirmation of the ischemic type of stroke, each patient was assessed by both physical examination and clinical investigations before enrolling into the study. Patients were excluded if they had any of the followings- a known or possible cardiac source of emboli or atrial fibrillation; having history of vascular disease, active infections, renal disease (estimated glomerular filtration rate <60 ml/min), liver disease, or thyroid dysfunction; history of Alzheimer's disease (AD), Huntington's disease (HD), Parkinson's disease (PD), or Multiple Sclerosis (MS); or under medications that are known to affect serum uric acid levels such as loop diuretics, salicylates, pyrazinamide, probenecid, ACE inhibitor, or benzbromarone. Same exclusion criteria were applied while selecting controls. For each case, one control was selected from patients attending the out-patient department who had no history of previous stroke and which was confirmed by clinical examinations. Controls were matched with the cases in terms of both age (±5 years) and sex.

## Data collection and variables analyzed

Data were collected administering a pre-tested questionnaire through face to face interviews. Content of the questionnaire included socio-demographic information like age, sex, educational status, and history of tobacco consumption.

Tobacco smoking was recorded as non-smoker (never smoked tobacco products), former smoker (had not smoked tobacco products in the past twelve months) and current smoker (smoking tobacco products on a daily or less than daily basis) [16]. Smoking status was then categorized as non-smoker and ever-smoker (former and current smoker) for analysis. Again, smokeless tobacco consumption was recorded in yes or no. Tobacco smoking and smokeless tobacco consumption were then combined to form a dichotomous composite variable of tobacco consumption (yes and no).

For laboratory variables, venous blood was drawn from each participant after an overnight fasting of at least 8 hours to measure fasting blood glucose, serum lipid profile, and SUA. For oral glucose tolerance test (OGTT), another blood sample was collected after 2 hours of 75 grams oral glucose load. Blood glucose was measured by glucose-oxidase method and rest of the measurements were assessed by standard enzymatic method using automated biochemical analyzer (Beckman Coulter AU480, Tokyo, Japan). All the tests were done in the mentioned hospital settings within a few hours of sample collection. The intra- and inter-assay coefficients of variation (CV) were within 5% and 10%, respectively.

A participant was considered having diabetes mellitus (DM) if diagnosed by OGTT (fasting blood glucose value $\geq$ 7.0 mmol/L and/or 2-h post-load glucose concentration $\geq$ 11.1 mmol/L) [17] or currently receiving treatment for DM.

SUA level was recorded in mg/dl and can be converted to μmol/L by multiplying with 59.485. Although consensus on the definition of hyperuricemia is scarce [18], serum uric acid levels $\geq$ 7 mg/dl in men and SUA $\geq$ 6.5 mg/dl in women was considered as hyperuricemia [19].

Dyslipidemia was diagnosed following Japan Atherosclerosis Society guideline [20] where a participant was considered having dyslipidemia if any of the following criteria was met–serum low density lipoprotein (LDL)-cholesterol ≥ 140 mg/dl or high density lipoprotein (HDL)-cholesterol < 40mg/dl or triglyceride ≥ 150mg/dl.

Participant was considered hypertensive if clinically diagnosed (systolic blood pressure (BP) ≥ 140 mm of Hg and/or diastolic BP ≥ 90 mm of Hg) or currently receiving medication for hypertension (HTN). Participant was considered having coronary heart disease (CHD) if one had the history of angina or infarction (ischemic heart disease) which was later confirmed by chest X-ray, ECG, and echocardiography.

### Ethical consideration

Ethical approval was obtained from the Ethical Review Board of Bangladesh Medical Research Council (BMRC/NRCE/2013-2016/637). Informed written consents were obtained from all participants prior to data collection after duly informing the objectives of this study.

### Statistical analysis

Qualitative variables were expressed by frequency and percentage, and quantitative variables by mean and standard deviation (SD). Comparison of distribution of categorical data among case and control group was done by Chi-square test and continuous data by independent t-test. Clinically relevant variables and variables found to be significant at 5% level in bivariate analysis were entered into multivariate analysis. Binary logistic regression analysis was done taking ischemic stroke as the outcome variable and age, sex, tobacco consumption, diabetes mellitus, hypertension, CHD, total cholesterol level, triglyceride level, and SUA level as exposure variables. Three regression models were constructed separately for male, female, and total population considering SUA level as continuous variable. All analysis was done using SPSS version 21 and p-value was considered significant at 5% level.

## Results

A total of 338 participants were recruited where 169 were cases and 169 were controls.

Characteristics of the study population are shown in Table 1. Among the participants, about 31 percent of cases were found to have hyperuricemia while only 14 percent of controls fell into this category. There was no significant difference in age, sex, and educational status between the case and control group. The case group comprised significantly more of tobacco consumer, diabetic, CHD, and hypertensive participants compared to the control group (p values <0.001). Dyslipidemia was observed significantly more among the cases (p<0.05). There was a significant difference in mean SUA level between cases (6.03 ± 1.84 mg/dl or 358.58 ± 109.31 μmol/L) and controls (4.34 ± 1.60 mg/dl or 258.27 ± 95.36 μmol/L).

Assessing SUA level in cases and controls revealed that SUA level was significantly higher among hypertensive ischemic stroke patients. There was no difference in SUA level in ischemic stroke patients with or without diabetes mellitus, CHD, dyslipidemia, or tobacco consumption history. Among the controls, significantly higher SUA level was found in participants having hypertension, diabetes mellitus, and CHD (Table 2).

After adjustment of age, sex, tobacco consumption, hypertension, diabetes mellitus, CHD, total cholesterol and triglyceride level, SUA level was found to be significantly associated with ischemic stroke (Table 3). For one-unit increase in SUA level, the odds of having ischemic stroke was increased by 25 percent (95% CI = 1.02–1.53; p<0.05). Analysis of SUA level separately for male and female showed a different picture. Elevated SUA level was found to be significantly associated with ischemic stroke among females (OR = 1.49; 95% CI = 1.01–2.19;

**Table 1.  Characteristics of study population (n = 338).**

| Variables | Group | | p value |
|---|---|---|---|
| | Case (n = 169) | Control (n = 169) | |
| Age (in year) [a] | 62.99 ± 12.19 | 62.88 ± 12.17 | 0.94 |
| Sex[o] | | | 1.00 |
| Male | 101 (59.80) | 100 (59.20) | |
| Female | 68 (40.20) | 69 (40.80) | |
| Education status[o] | | | 0.27 |
| Up to secondary | 101 (59.80) | 90 (53.30) | |
| Above secondary | 68 (40.20) | 79 (46.70) | |
| Smoking Status[o] | | | 0.000* |
| Ever smoker | 75 (63.60) | 43 (36.40) | |
| Non-Smoker | 94 (42.70) | 126 (57.30) | |
| Diabetes Mellitus[o] | 79 (46.70) | 15 (8.90) | 0.000* |
| CHD[o] | 55 (32.50) | 16 (9.50) | 0.000* |
| Hypertension[o] | 109 (64.50) | 19 (11.20) | 0.000* |
| Dyslipidemia[o] | 159 (94.10) | 103 (60.90) | 0.000* |
| Fasting blood sugar (mmol/L) [a] | 7.91 ± 3.79 | 6.24 ± 2.75 | 0.000* |
| Total cholesterol (mg/dl) [a] | 187.25 ± 47.85 | 143.08 ± 31.46 | 0.000* |
| LDL cholesterol (mg/dl) [a] | 161.21 ± 34.27 | 103.78 ± 21.75 | 0.000* |
| HDL cholesterol (mg/dl) [a] | 40.76 ± 5.89 | 40.28 ± 6.42 | 0.475 |
| Triglyceride (mg/dl) [a] | 186.78 ± 111.37 | 151.34 ± 70.09 | 0.001* |
| Serum Uric Acid (mg/dl)[a] | 6.03 ± 1.84 | 4.34 ± 1.60 | 0.000* |

[o]p value was derived by Chi-square test and

[a] Independent t-test was applied

p<0.05) but not in males. Other statistically significant predictors of ischemic stroke among the participants were found to be hypertension, diabetes mellitus, tobacco consumption, and high total cholesterol level.

## Discussion

Significantly elevated SUA level is found in ischemic stroke patients compared to controls after controlling the effect of age, sex, and other cardiovascular risk factors such as hypertension, diabetes, CHD, total cholesterol level, triglyceride level, and tobacco consumption. Positive predictive role of SUA in acute ischemic stroke has been found in several studies conducted in different countries [2,21,22]. However, the exact pathophysiological role of SUA in ischemic stroke remains little known [23].

From the evidences till now, some hypotheses can be made to describe the association between SUA and ischemic stroke. Firstly, elevated SUA level in ischemic stroke reflects the accumulation of other cardiovascular risk factors such as hypertension, diabetes, metabolic disorder, atrial fibrillation etc. [9]. In accordance with this hypothesis, if elevated SUA level in ischemic stroke merely reflects its association with cardiovascular risk factors, it is to be expected that this relation will be attenuated when these risk factors are adjusted for. However, this study finding shows that elevated SUA level is significantly associated with ischemic stroke even after controlling the effect of other cardiovascular risk factors. Moreover, there was no significant difference in SUA level in ischemic stroke patients with or without diabetes, CHD or dyslipidemia. Hence, clustering of risk factors does not fully explain the positive relation

**Table 2. Distribution of serum uric acid level in cases and controls with different risk factors.**

| Risk factors | Serum Uric acid Mean (SD) | |
|---|---|---|
| | Cases | Controls |
| **Tobacco consumption** | | |
| Yes | 5.99 (1.92) | 4.56 (1.69) |
| No | 6.13 (1.54) | 4.08 (1.46) |
| p-value | 0.654 | 0.051 |
| **Diabetes mellitus** | | |
| Yes | 6.09 (1.97) | 5.86 (2.21) |
| No | 5.97 (1.72) | 4.19 (1.46) |
| p-value | 0.682 | 0.000 |
| **Hypertension** | | |
| Yes | 6.31 (1.58) | 5.59 (1.83) |
| No | 5.52 (1.91) | 4.18 (1.51) |
| p-value | 0.007 | 0.000 |
| **CHD** | | |
| Yes | 6.32 (2.03) | 6.38 (1.85) |
| No | 5.89 (1.73) | 4.13 (1.42) |
| p-value | 0.153 | 0.000 |
| **Dyslipidemia** | | |
| Yes | 6.07 (1.84) | 4.44 (1.65) |
| No | 5.35 (1.80) | 4.18 (1.53) |
| p-value | 0.229 | 0.306 |

between stroke and uric acid, and this study extrapolated an independent association between them.

Secondly, elevated SUA level is associated with endothelial dysfunction and increased platelet adhesiveness which predispose to thrombus formation, elevated circulating levels of

**Table 3. Logistic regression result predicting likelihood of presenting with ischemic stroke both for male and female.**

| Variables | Male [a] | Female [b] | Both [c] |
|---|---|---|---|
| | OR (95% CI) | OR (95% CI) | OR (95% CI) |
| Age | 1.02 (0.99–1.06) | 1.02 (0.97–1.08) | 1.02 (0.99–1.05) |
| Sex | - | - | 1.16 (0.57–2.34) |
| Tobacco Smoking | 3.43 (1.31–9.02) * | 2.21 (0.71–6.88) | 2.67 (1.33–5.35) * |
| Diabetes | 6.39 (2.04–20.01) * | 17.20 (3.42–86.44) * | 7.69 (3.28–18.08) ** |
| Hypertension | 16.45 (5.14–52.69) ** | 4.58 (1.32–15.88) * | 11.01 (4.93–24.59) ** |
| CHD | 0.53 (0.17–1.69) | 0.12 (0.01–1.61) | 0.49 (0.19–1.26) |
| Total cholesterol level | 1.02 (1.01–1.03) * | 1.03 (1.02–1.05) ** | 1.02 (1.01–1.03) ** |
| Triglyceride level | 1.00 (0.99–1.01) | 1.00 (0.99–1.01) | 1.00 (0.99–1.00) |
| SUA level | 1.22 (0.94–1.59) | 1.49 (1.01–2.19) * | 1.25 (1.02–1.53) * |

[a]Cox & Snell $R^2$ = 0.451 and Nagelkerke $R^2$ = 0.601

[b]Cox & Snell $R^2$ = 0.465 and Nagelkerke $R^2$ = 0.621

[c]Cox & Snell $R^2$ = 0.449 and Nagelkerke $R^2$ = 0.599

*p-value less than 0.05 and

** p-value less than 0.001

systemic inflammatory mediators, and vascular smooth muscle proliferation [11]. Thus, hyperuricemia exerts a pathological role in development of arterial stiffness [24,25], atherosclerosis and essential hypertension [26,27] and leads to ischemic stroke. Partially consistent with this hypothesis, in this study, a significantly higher level of SUA was found in hypertensive ischemic stroke patients comparing to the non-hypertensive one. However, the incidence of ischemic stroke in hyperuricemic subjects without hypertension cannot be explained by this hypothesis and therefore needs further exploration.

Thirdly, free radical scavenger effect of uric acid results in elevation of SUA level [28] after an ischemic stroke incident. The time of measuring SUA levels is an important factor to be looked into as there is a fluctuating pattern of uric acid level evident in ischemic stroke patients. Ji Man Hong [29] reported that SUA level falls initially after onset of stroke symptoms and then raises gradually over seven days. Studies reporting positive predictive role of SUA level in ischemic stroke mostly measured SUA level immediately after an incident [30]. In the present study, SUA level in cases was measured at the first morning after admission no more than three days after an ischemic event. In contrast, a study by Sridharan [31], where SUA was measured at least seven days after an acute stroke, concluded that low level of SUA is a risk factor of ischemic stroke. Thus, it calls for further prospective study on SUA and ischemic stroke (e.g. relation between prior hyperuricemia and incident stroke) before concluding elevated SUA level as a risk factor of ischemic stroke.

This study analyzed the role of SUA in ischemic stroke separately for both male and female. Although the elevated level of SUA significantly increases the odds of having ischemic stroke among females, it losts its significance in case of males. Many studies reported a stronger association of hyperuricemia with stroke in female [32]. A Taiwanese study reported that the risk of ischemic stroke is more among female with hyperuricemia compared to their male counterpart after adjusting multiple risk factors (HR 1.32; 95% CI: 1.00–1.73) [33]. In contrary, a Swedish prospective study called Apolipoprotein Mortality Risk study (AMORIS) described a stronger relationship of uric acid level with stroke in men compared to women [1]. In a cohort of female, SUA level was reported not to be associated with ischemic stroke after adjustment of cardiovascular risk factors [34]. Yet, plausible mechanism for gender-specific predictive role of SUA remains under-reported.

It should be noted however, that a considerable amount of studies did not find any relation of SUA with ischemic stroke [35]. Many studies failed to establish this association while considering the effect of hypertension, diabetes, or dyslipidemia [36]. On the other hand, there is accumulating evidence that uric acid plays a free radical scavenger role in cerebral injury suggesting administration of uric acid in acute phase of stroke to minimize neurological deficits [9,10].

This study was based on data collected from a department at a tertiary care hospital in Dhaka, making it less representative for the entire population. Besides, population recruited in the control group was also taken from hospital patients that might lead to selection bias. Although cases of this study were directly supervised for overnight fasting before collecting blood samples, supervision of the controls was not possible as they were taken from the outpatient department. Some other variables known to influence SUA level such as body mass index (BMI) and metabolic disorder are not addressed in this study. As majority of the cases participated in this study were bed-ridden, it was not possible to measure weight and height for all participants and thus lacking data on BMI. However, this study provides a unique perspective since it has analyzed the role of SUA in ischemic stroke while considering some well-established cardiovascular risk factors, along with an analysis of the outcome for both male and female.

## Conclusion

This study inferred that elevated level of SUA is significantly associated with the acute phase of ischemic stroke while controlling the effect of cardiovascular risk factors like hypertension, diabetes, CHD, total cholesterol level, triglyceride level, and tobacco consumption. In gender-specific analysis, presence of higher level of SUA was found only in female ischemic stroke patients. This study calls for future prospective study to generate more scientific evidence on role of elevated SUA levels in ischemic stroke among Bangladeshi population.

## Acknowledgments

Authors want to acknowledge Dr. Mahmood-Uz-Jahan and Prof. Ridwanur Rahman for their formative criticism, and Muhammad Ibrahim Ibne Towhid and M. Takit Mallik for their contribution in manuscript review.

## Author Contributions

**Conceptualization:** Mohammad Ibrahim Khalil, Marium Salwa, Mohammad Abdullah Al Mamun, M. Atiqul Haque.

**Data curation:** Marium Salwa, Nilima Barman.

**Formal analysis:** Mohammad Ibrahim Khalil, Marium Salwa, Sarmin Sultana, Nilima Barman.

**Investigation:** Mohammad Ibrahim Khalil, Marium Salwa, Mohammad Abdullah Al Mamun, Nilima Barman, M. Atiqul Haque.

**Methodology:** Mohammad Ibrahim Khalil, Marium Salwa.

**Project administration:** Mohammad Ibrahim Khalil, Mohammad Abdullah Al Mamun, M. Atiqul Haque.

**Resources:** Mohammad Ibrahim Khalil.

**Software:** Marium Salwa.

**Supervision:** Mohammad Ibrahim Khalil, Marium Salwa, Mohammad Abdullah Al Mamun, M. Atiqul Haque.

**Visualization:** Mohammad Ibrahim Khalil.

**Writing – original draft:** Mohammad Ibrahim Khalil, Marium Salwa, Sarmin Sultana.

**Writing – review & editing:** Mohammad Ibrahim Khalil, Marium Salwa, Sarmin Sultana, Mohammad Abdullah Al Mamun, Nilima Barman, M. Atiqul Haque.

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
