## [Decision Letter · Decision Letter 0]

6 May 2020

PONE-D-20-03190

Role of serum uric acid in ischemic stroke: A case-control study in Bangladesh

PLOS ONE

Dear Dr. Salwa,

Thank you for submitting your manuscript to PLOS ONE. After careful consideration, we feel that it has merit but does not fully meet PLOS ONE’s publication criteria as it currently stands. Therefore, we invite you to submit a revised version of the manuscript that addresses the points raised during the review process.

We would appreciate receiving your revised manuscript by Jun 20 2020 11:59PM. To enhance the reproducibility of your results, we recommend that if applicable you deposit your laboratory protocols in protocols.io, where a protocol can be assigned its own identifier (DOI) such that it can be cited independently in the future. For instructions see: http://journals.plos.org/plosone/s/submission-guidelines#loc-laboratory-protocols

We look forward to receiving your revised manuscript.

Kind regards,

Marietta Zille, PhD

Academic Editor

PLOS ONE

**Comments to the Author**

Reviewer #1: The proposed paper is interesting and well-written. However some revision are needed before it could be accepted for pubblication:

- Although the use of categorical variables permit to obtain OR the use of continuous variables seems to be more usefull for multivariate model. Please also add a model (logistic regressione) with stroke has the dependent variables and with age, systolic BP, glucose, total cholesterol and uric acid as covariates and provide the relative beta for the association. Since authors describe in the methods section that they calculated the relative categorical variable (hypertension, dyslipidemia and so on) from the relative continue variables those are surely presents and usable for the analysis. Furthermore continue values need to be added to table 1.

- Why Atrial Fibrillation falls under the definition of CHD? if it is a mistake please correct.

- When discussing on gender role in relatioship between uric acid and stroke or even with CV events in general please also cite two recent pubblication on this issue (i.e. Association between uric acid and pulse wave velocity in hypertensive patients and in the general population: a systematic review and meta-analysis. Blood Press. 2020 [Epub ahead of print] and Pulse wave velocity progression over a medium-term follow-up in hypertensives: Focus on uric acid. J Clin Hypertens (Greenwich). 2019 Jul;21(7):975-983).

- One minor comments: please don't use the word "attack" through the Whole text since it doesn0t sound very scientific.

Reviewer #2: Interesting paper

Sme issues should be added

Abstract, It should be written again dividing into introduction/methods/results/conclusion

Methods. N of patients should not written in the methods but in the results.

Methods. Due to reduced sample size normal distribution should be checked for

Methods. Level of significance for choosing variables in the multivariate analysis should be added.

Reviewer #3: The authors of the present manuscript aim to evaluate the role of serum uric acid (SUA) in patients with ischemic stroke, assessing sex-based differences and try to differentiate it from other cardiovascular risk factors. They carried out a case control study, being the cases patients with a diagnosis of ischemic stroke and the controls, outpatients attended to the neurology department without history of previous stroke. The final conclusion is that elevated SUA level is a significant risk factor for ischemic stroke with an association exclusively in males.

The purpose of the study is interesting. However, it does not add any new relevant information related to the topic. Additionally some questions should be elucidated according to STROBE statement.

- Title and abstract provide an informative and balanced summary of what was done and what was found.

-Introduction is good. However, a prespecified hypotheses is lacking.

-Methods:

– - The authors compare a group (cases) whose have been admitted into hospital because of ischemic stroke with other group (controls) whose were outpatients with different neurologic disorders. Probably, the different setting (in-patients vs outpatients) in the groups introduces a bias in selection because of the fact that it has been described higher SUA levels in the acute phase of stroke. On the other hand, it has been also reported higher SUA levels in other neurologic disorders such as Parkinson disease or multiple sclerosis what make more difficult to interpret the results.

– - In this regard, the authors shoud explain how matching of cases and controls was addressed.

– - Some important variables which could influence the results are lacking. The authors should add information about body mass index (the presence of obesity and metabolic syndrome are strongly related to SUA levels), renal disease or glomerular filtration rate, and they should add information regarding the treatment of the both samples. We do not know if there were patients under uricosuric drugs for instance. It is known that some drugs apart from loop diuretics or thiazides are able to modify the levels of SUA (ACEi, ARB.)

– - The authors should explain how the study size was arrived at.

- Results: The main results are well described and reported.

- Discussion: The main finding of the present study is that the cases of ischemic stroke had higher levels of SUA than those of the controls. From that point on, the authors assume a predictive positive role of the uric acid in the incidence of ischemic stroke what has not been able to infere from the design of the present study, therefore the discussion and the conclussion are confusing and they must be changed in order to explain accurately the real interpretation of the results.

---

## [Author Response · Author response to Decision Letter 0]

19 Jun 2020

Reviewer #1

1. Although the use of categorical variables permits to obtain OR the use of continuous variables seems to be more useful for multivariate model. Please also add a model (logistic regression) with stroke has the dependent variables and with age, systolic BP, glucose, total cholesterol, and uric acid as covariates and provide the relative beta for the association. Since authors describe in the methods section that they calculated the relative categorical variable (hypertension, dyslipidemia and so on) from the relative continue variables those are surely presents and usable for the analysis. 

Response: Regression model has been re-constructed according to the suggestion. Page 12 (Table 3).

As many of our participants were on antihypertensive and anti-diabetic medication, their blood pressure and OGTT measurement during data collection did not accurately reflect their hypertensive or diabetic status. Thus, including systolic BP and blood glucose level in regression analysis might not exactly represent their association with ischemic stroke. So, we have included total cholesterol level and triglyceride level to represent the lipid profile of the participants while excluding categorical variable of dyslipidemia from the regression model as per your suggestion. Result of the study has been changed due to the change in nature of variables. 

*A regression model is attached below constructed according to the reviewer’s instruction. 

2. Furthermore, continue values need to be added to table 1

Response: Added Page 9 (Table 1)

3. Why Atrial Fibrillation falls under the definition of CHD? if it is a mistake please correct 

Response: Corrected Page 8, Line 148

4. When discussing on gender role in relationship between uric acid and stroke or even with CV events in general please also cite two recent publication on this issue (i.e. Association between uric acid and pulse wave velocity in hypertensive patients and in the general population: a systematic review and meta-analysis. Blood Press. 2020 [Epub ahead of print] and Pulse wave velocity progression over a medium-term follow-up in hypertensives: Focus on uric acid. J Clin Hypertens (Greenwich). 2019 Jul;21(7):975-983). 

Response: Addressed Page 14, Line 224

5. - One minor comments: please don't use the word "attack" through the Whole text since it doesn0t sound very scientific. Response: Edited All over the text 

*Table. Logistic regression result showing association of different biochemical and physical measurement with acute ischemic stroke (Requested by the reviewer #1)

Variables Male a Female b Both c

 OR (95% CI) OR (95% CI) OR (95% CI)

Age 1.027 (0.99-1.06) 1.05 (0.99- 1.09) 1.03 (1.01-1.05)

Sex - - 

TC 1.021 (1.01-1.03)* 1.03 (1.02- 1.05)* 1.025 (1.02-1.03)

Fasting blood glucose 1.13 (1.02-1.27)* 1.25 (0.99-1.58) 1.129 (1.03-1.24)

Systolic BP 1.04 (1.03-1.05)* 1.03 (1.01- 1.05)* 1.036 (1.03-1.05)

SUA level 1.25 (0.98-1.60) 1.69 (1.15-2.50)* 1.318 (1.09-1.60)

*P-value < 0.05 

Reviewer’s comment Response

Reviewer #2 

1. Abstract, it should be written again dividing into introduction/methods/results/conclusion 

Response: Modified Page 2, Line 24-44

2. Methods. N of patients should not write in the methods but in the results. 

Response: Corrected Page 7, line 114 and Page 9, line 166

3. Methods. Due to reduced sample size normal distribution should be checked for 

Response: Continuous variables were normally distributed. 

4. Methods. Level of significance for choosing variables in the multivariate analysis should be added. 

Response: Corrected Page 9, line 158

Reviewer’s comment Response

Reviewer #3

1. Introduction is good. However, a prespecified hypotheses is lacking. 

Response: Research questions are added Page 5

2. Methods:

– - The authors compare a group (cases) whose have been admitted into hospital because of ischemic stroke with other group (controls) whose were outpatients with different neurologic disorders. Probably, the different setting (in-patients vs outpatients) in the groups introduces a bias in selection because of the fact that it has been described higher SUA levels in the acute phase of stroke. On the other hand, it has been also reported higher SUA levels in other neurologic disorders such as Parkinson disease or multiple sclerosis what make more difficult to interpret the results.

– - In this regard, the authors should explain how matching of cases and controls was addressed.

Response: Selection and matching procedures have been elaborated under a separate heading “Cases and controls” Page 5

2. – - Some important variables which could influence the results are lacking. The authors should add information about body mass index (the presence of obesity and metabolic syndrome are strongly related to SUA levels), renal disease or glomerular filtration rate, and they should add information regarding the treatment of the both samples. We do not know if there were patients under uricosuric drugs for instance. It is known that some drugs apart from loop diuretics or thiazides are able to modify the levels of SUA (ACEi, ARB.)

Response: - The list of drugs used during data collection has been provided. Page 6, line 97-99

- We were not able to measure height and weight for all the participants as many of them were bed-ridden, thus lacking BMI in the study outcome. It has been addressed in the limitation. Page 16, line 267-270

- Participants with glomerular filtration rate less than 60 ml/sec were excluded initially from the study. Page 6, line 95

3. The authors should explain how the study size was arrived at. 

Response: We employed consecutive sampling to enroll the patients for both case and control groups within our data collection period of six months. Therefore, we did not go for formal sample size calculation. Page 5, Line 81-85

4. Discussion: The main finding of the present study is that the cases of ischemic stroke had higher levels of SUA than those of the controls. From that point on, the authors assume a predictive positive role of the uric acid in the incidence of ischemic stroke what has not been able to infer from the design of the present study, therefore the discussion and the conclusion are confusing and they must be changed in order to explain accurately the real interpretation of the results. 

Response: Corrected accordingly. Throughout the whole text

---

## [Decision Letter · Decision Letter 1]

14 Jul 2020

Role of serum uric acid in ischemic stroke: A case-control study in Bangladesh

PONE-D-20-03190R1

Dear Dr. Salwa,

We’re pleased to inform you that your manuscript has been judged scientifically suitable for publication and will be formally accepted for publication once it meets all outstanding technical requirements.

Kind regards,

Marietta Zille, PhD

Academic Editor

PLOS ONE

---

## [Editor Report · Acceptance letter]

23 Jul 2020

PONE-D-20-03190R1 

Role of serum uric acid in ischemic stroke: A case-control study in Bangladesh 

Dear Dr. Salwa:

I'm pleased to inform you that your manuscript has been deemed suitable for publication in PLOS ONE. Congratulations! Your manuscript is now with our production department. 

Kind regards, 

on behalf of

Dr. Marietta Zille 

Academic Editor

PLOS ONE